# Boosting superconductivity in ultrathin YBa$_2$Cu$_3$O$_{7-\delta}$ films via nanofaceted substrates

Eric Wahlberg [1,2], Riccardo Arpaia [1,3], Debmalya Chakraborty [4,5,6], Alexei Kalaboukhov[1], David Vignolles[7], Cyril Proust [7], Annica M. Black-Schaffer [6], Thilo Bauch[1], Götz Seibold [8] ✉ & Floriana Lombardi [1] ✉

In cuprate high-temperature superconductors the doping level is fixed during synthesis, hence the charge carrier density per CuO$_2$ plane cannot be easily tuned by conventional gating, unlike in 2D materials. Strain engineering has recently emerged as a powerful tuning knob for manipulating the properties of cuprates, in particular charge and spin orders, and their delicate interplay with superconductivity. In thin films, additional tunability can be introduced by the substrate surface morphology, particularly nanofacets formed by substrate surface reconstruction. Here we show a remarkable enhancement of the superconducting onset temperature $T_c^{on}$ and the upper critical magnetic field $H_{c,2}$ in nanometer-thin YBa$_2$Cu$_3$O$_{7-\delta}$ films grown on a substrate with a nanofaceted surface. We theoretically show that the enhancement is driven by electronic nematicity and unidirectional charge density waves, where both elements are captured by an additional effective potential at the interface between the film and the uniquely textured substrate. Our findings show a new paradigm in which substrate engineering can effectively enhance the superconducting properties of cuprates. This approach opens an exciting frontier in the design and optimization of high-performance superconducting materials.

Cuprates are notable for their strong electron-electron interactions, which are believed to drive both their high critical temperature and the emergence of various correlated electronic phases, including the charge density wave (CDW) phase that competes with superconductivity[1–8]. However, understanding these interactions theoretically remains difficult. Additional complexities arise from the intricate stoichiometry of cuprates, which complicates the distinction of doping effects from structural changes, limiting progress in understanding why their critical temperature surpasses that of conventional superconductors.

Recently, uniaxial strain tuning has emerged as a powerful method for modifying the properties of cuprates and other strongly correlated systems[9–17]. This technique has proven particularly effective in altering the CDW order[18–23]. In YBa$_2$Cu$_3$O$_{7-\delta}$ (YBCO), applying uni-axial strain that compresses the $b$-axis enhances CDW along the $a$-axis, while leaving it mostly unchanged along the $b$-axis. The vice versa also

[1]Quantum Device Physics Laboratory, Department of Microtechnology and Nanoscience, Chalmers University of Technology, Göteborg, Sweden. [2]RISE Research Institutes of Sweden, Borås, Sweden. [3]Department of Molecular Sciences and Nanosystems, Ca' Foscari University of Venice, Venice, Italy. [4]Department of Physics, Birla Institute of Technology and Science—Pilani, K. K. Birla Goa Campus, Sancoale, India. [5]Department of Physical Sciences, Indian Institute of Science Education and Research (IISER) Mohali, Manauli, India. [6]Department of Physics and Astronomy, Uppsala University, Uppsala, Sweden. [7]LNCMI-EMFL, CNRS UPR3228, Université Grenoble Alpes, Université de Toulouse, INSA-T, Toulouse, France. [8]Institut für Physik, BTU Cottbus-Senftenberg, Cottbus, Germany. ✉e-mail: goetz.seibold@b-tu.de; floriana.lombardi@chalmers.se

applies. However, while $b$-axis compression has only a minor effect on the superconducting transition temperature $T_c$, compression along the $a$-axis leads to a steep decrease in $T_c$. The origin of this behavior remains an open question.

In striped materials such as $La_{2-x}Sr_xCuO_4$ (LSCO), which display spin-charge separation, even minimal uniaxial stress leads to a notable reduction in magnetic volume fraction and a substantial enhancement in the onset of 3D superconductivity along the $c$-axis[10,22]. At high enough strain, the onset of 2D superconductivity fully aligns with the 3D transition onset, making the sample effectively 3D.

In all these studies, uniaxial compressive strain is applied to single crystals. While this strain significantly influences the charge and spin order, the resulting $T_c$ is either suppressed or does not exceed the maximum bulk value achievable for the specific system.

In thin films, the mismatch with the substrate can introduce tensile or compressive strains that often surpass those achievable in single crystals under uniaxial pressure. This dual ability to stretch and compress the unit cell significantly expands substrate strain tunability. Additionally, the substrate's unique morphology, such as nanofacets formed during high-temperature deposition, can alter the material properties beyond simple strain effects, offering new possibilities for material engineering.

In recent works, we have demonstrated that the normal state properties of nm-thick films of YBCO are significantly influenced by a combination of tensile strain and unidirectional faceting formed on the (110)-oriented surface of MgO substrates[24,25] (see Fig. 1(a)). When the YBCO film is deposited, the $CuO_2$ layers interact with the MgO substrate through a hopping parameter, $t_\perp$, mainly at the nanofacet ridge tips formed by high-temperature surface reconstruction before deposition (see Fig. 1(a)). This means that only specific elongated sections of the $CuO_2$ layer (along the $b$-axis) gain this extra $t_\perp$ coupling. When charge carriers move virtually to the substrate atoms, they create a repulsion between the energy levels of the film and the substrate, producing an effective potential $V_{eff}$ for the YBCO atoms close to the interface[25], see Supplementary Section D. We have demonstrated that this model is able to predict the presence of electronic nematicity, as we will describe below, and a unidirectional CDW in nm-thick films. This is contrary to bulk crystals, where the CDW is bidirectional. Due to screening, the effect of this potential is more pronounced in ultrathin films than in thicker ones.

Nematicity and unidirectional CDW, induced by nanofaceted substrates, raise a natural question: Can nanofaceting also affect the material's superconducting properties? We provide a positive answer, demonstrating a significant enhancement in $T_c$ and the upper critical magnetic field $H_{c,2}$: 10 nm films exhibit a $T_c$ over 15 K higher and an $H_{c,2}$ more than 50 T greater than in 50 nm films. We argue that these remarkable effects result from the induced electronic order, captured by the $V_{eff}$ potential acting on the $CuO_2$ planes. Thus, substrate engineering offers a new avenue to enhance cuprate superconductivity.

## Results

### Anisotropic transport in strained films

The YBCO films used in this study are grown on (110) oriented MgO substrates following the procedure described in refs. [26–28]. The films span a wide range of hole-doping $p$, going from the strongly underdoped ($p \approx 0.06$) up to the slightly overdoped ($p \approx 0.18$) regime and for thicknesses $d$ in the range 10-50 nm. Measurements of the $a-$, $b-$, and $c-$axis lengths indicate a tensile strain increasing with the reduction of the thickness[24]. The substrates are annealed at high temperature ($T = 790°C$) before the film deposition to reconstruct the substrate surface. This procedure promotes the formation of triangular nanofacets on the surface of the MgO with average height of 1 nm and width in the range 20-50 nm and consequently the growth of untwinned YBCO films[28].

The temperature dependence of the resistivity along the YBCO $a$- and $b$-axis for 50 and 10 nm is shown in Fig. 1(b) and 1(c), respectively. Each film has been measured in a four-point Van der Pauw configuration[29] in a PPMS cryostat with a 14 T magnet. The results of the Van der Pauw measurements have been confirmed by resistivity measurements of patterned Hall bars aligned along the $a$- and $b$-axis (see refs. [27,30,31] for details on the patterning procedure).

The in-plane resistivity is strongly anisotropic in the 10 nm thick films (see Fig. 1(c)) compared to 50 nm thick films (see Fig. 1(b)) with the same doping level $p = 0.125$. We observe that the slope of the $\rho(T)$ is very different between the $a$ and the $b$ directions, and that the resistivity anisotropy $\rho_a/\rho_b$ is strongly enhanced even at room temperature. These properties can be accounted for by considering that the Fermi surface becomes nematic in very thin films[24]. This conclusion comes after establishing that the two films indeed have the same doping.

### Determination of hole doping in 10 nm films

When comparing the doping between $d = 10$ nm and $d = 50$ nm thick films we rely on the assumption that an empirical parabolic $T_c(p)$ dependence is valid for $p < 0.08$ and $p > 0.15$, and that the $c$-axis changes monotonically with doping[27]. The doping level can be determined by x-ray diffraction measurements of the YBCO $c$-axis length together with the measured $T_c^{on}$ at 90% of the normal state resistivity, as described elsewhere[27,32] (see also Supplementary Section A). This procedure allows to establish the doping $p = 0.125$ for the 50 nm film in Fig. 1(b). In this case, the pseudogap temperature $T^\star$, defined as the temperature where the resistivity deviates from linearity, which is equivalent to $T_L^a$ shown in Fig. 1(b) and 1(c), has the same value as single crystals for the same doping, and can therefore be considered a good reference measurement of the doping of the film. For the 10 nm thin films, we have that $T_c^{on}$ is 15 K higher, which would point to a much higher doping level, that however, does not align with the value of the $c$-axis that smoothly increases from the value at optimal doping. In addition, the measurement of the $T^\star$ value (see Fig. 1(c)) is very close to the 50 nm thin films, suggesting also the same level of doping. Figure 2 shows the evolution of $T_c^{on}$ as a function of $p$ for 10 nm thin films. The $T_c^{on}$ is improved for $0.12 < p < 0.15$ compared to our thicker films (red dash-dotted line)[27]. We find that the "ideal" parabolic superconducting dome is recovered down to $p \approx 0.12$. To further support that the 10 nm and 50 nm films in Fig. 2 have the same doping level, we have extracted $T^\star$ from the 10 nm films as a function of doping (see Supplementary Fig. 1(a)). Using linear interpolation, we have determined the $T^\star(p)$ dependence, enabling the reconstruction of the superconducting dome from $T^\star$ measurements instead of the $c$-axis (see Supplementary Fig. 1(b)). The striking similarity between the two domes confirms that both $T^\star$ and the $c$-axis serve as reliable indicators of the film's doping level.

For comparison, we have also deposited 10 nm and 50 nm thick YBCO (close to the 1/8 doping) on double-terminated (SrO and $TiO_2$) $SrTiO_3$ (STO) substrates. Here, the substrate surface does not undergo a reconstruction at high temperature (see Supplementary Fig. 2). We observe, for the same value of doping as Fig. 1(b)-(c), that both 10 nm and 50 nm have the same value of $T_c^{on}$, $c$-axis length and $T^\star$. There is therefore no increase in the $T_c^{on}$ on 10 nm films grown on (001) STO.

### Enhanced $T_c^{on}$ and CDW order

We have therefore arrived to the first important observation: 10 nm thick films grown on a nanofaceted surface show an increase of $T_c^{on}$ in the range $0.125 < p < 0.15$ that at 1/8 doping also exceeds the expected value for the ideal parabolic behaviour (see Fig. 2). Here, it is worth emphasizing that the observed increase in $T_c^{on}$ cannot be explained by, nor directly compared to, the enhancement seen in hydrostatic pressure experiments, where $T_c$ increases as a result of a homogeneous and simultaneous compression of the unit cell along all crystallographic

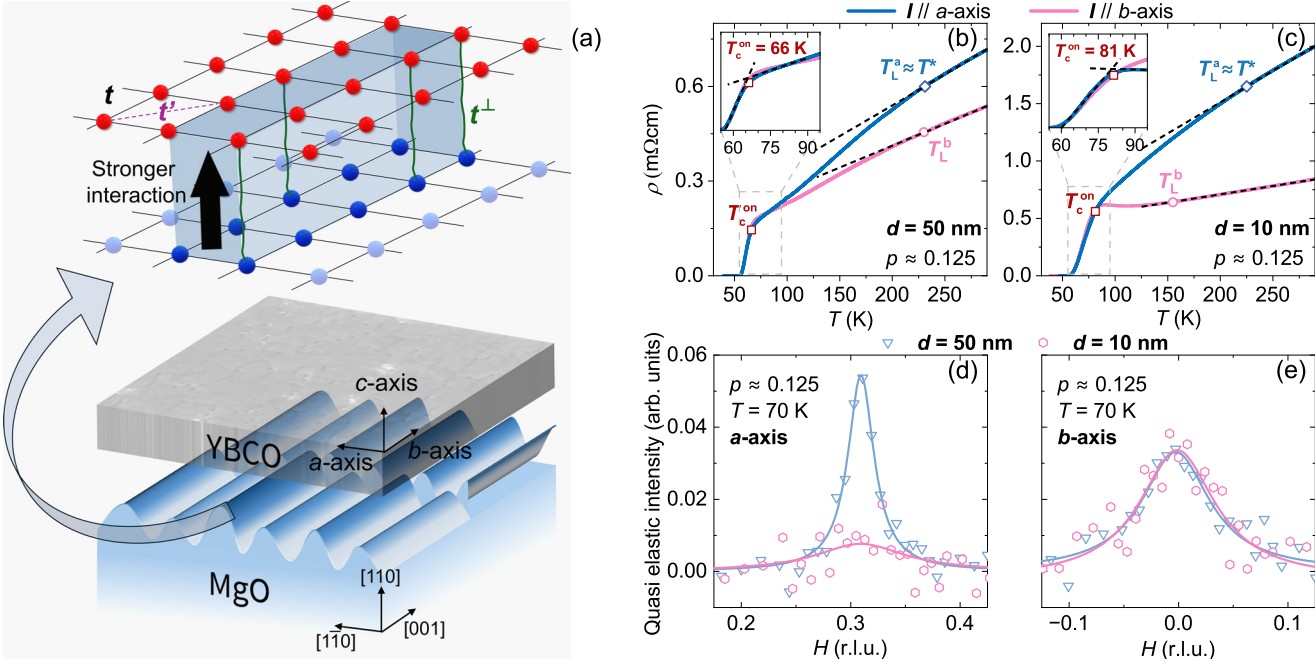

**Fig. 1 | Engineering the film-substrate interface: The origin of dramatic modifications in the YBCO ground state. a** By annealing (110)-oriented MgO substrates, nanoscale facets are created on the surface, leading to significant differences in atomic coordination between the valleys and the facet edges, particularly at the apexes (shown in the gradient of blue, with stronger color indicating lower coordination of surface atoms). When the YBCO film (grey slab) is deposited, it interacts with this anisotropic substrate matrix, since the YBCO unit cell height is comparable to the facet height. This induces strong coupling between the film and the substrate, mainly at the under-coordinated apex regions (darker blue), where hybridization occurs to saturate dangling bonds. The zoom-in on top illustrates the YBCO planar tight-binding structure at the atomic scale (red dots), with hopping parameters $t$ (nearest neighbor) and $t'$ (next nearest neighbor). The coupling between the under-coordinated substrate regions (dark blue dots) and the YBCO atoms is mediated by a coupling parameter $t^\perp$, while the coupling to the more

coordinated regions (light blue dots) is negligible. This is evident in both transport and spectroscopic measurements. Resistivity $\rho(T)$ measured along the $a$- (blue line) and $b$-axis (pink line) for a 50 nm thick film (**b**) and a 10 nm thick film (**c**). The black dashed lines are linear fits to the high-$T$ resistivity, while $T = T_L^{a,b}$ (blue diamonds and pink circles) mark the temperatures at which the linear $\rho(T)$ ends. In the 10 nm film, two key effects are observed: on one hand, the linear-in-$T$ resistivity along the $b$-axis extends to much lower temperatures compared to the $a$-axis, as previously discussed in ref. 24; second, the onset of the resistive transition $T_c^{on}$, defined at 90% of the normal state resistivity, is approximately 15 K higher in both in-plane directions compared to the $d = 50$ nm film. CDW signal, measured using Resonant Inelastic X-ray Scattering (RIXS) at $T = T_c = 70$ K for the $a$-axis (**d**) and $b$-axis (**e**). For each direction, the 50 nm film (triangles) and the 10 nm film (hexagons) are compared. The CDW is thickness independent along the $b$-axis, whereas it disappears along the $a$-axis for the 10 nm film.

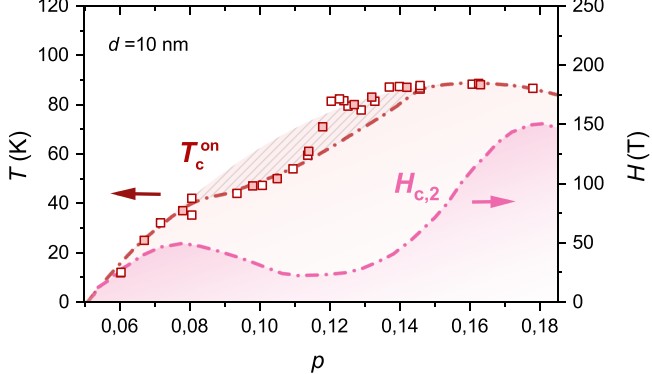

**Fig. 2 | Phase diagram of strained YBCO thin films.** Open and filled symbols are from measurements of unpatterned thin films and patterned Hall bars, respectively. All lines are guides for the eye. $T_c^{on}$ (red squares) is the superconducting transition onset temperature as defined in the main text. For comparison the red dash-dotted line shows $T_c^{on}$ measured in relaxed YBCO thin films and crystals and the red striped area the suppression of $T_c^{on}$ from quadratic $p$ dependence due to competition with the CDW order. $H_{c,2}$ is the upper critical field in YBCO single crystals.

directions[33,34]. In contrast, our situation is rather different: in the 10 nm-thick films, the unit cell is subject to a uniaxial tensile strain, imposed by the anisotropic epitaxial interface, in addition to a substrate potential induced by the nanofacets.

What is the origin of the remarkable enhancement of $T_c^{on}$ in 10 nm YBCO film grown on (110) MgO? Can it be attributed to a change in the CDW, notoriously competing with superconductivity?

In 10 nm thick YBCO films with hole doping $p \approx 0.125$, the CDW order is modified: it is completely suppressed along the $a$-axis while remaining unaltered along the $b$-axis (see pink hexagons in Figs. 1(d) and 1(e)), which makes the charge order unidirectional. This is in contrast to a reference 50 nm thick YBCO film where the CDW remains bidirectional, as expected for single crystals at this level of doping[35]. The weight of the CDW order is close to that measured in the 50 nm thick film along the $b$-axis (see triangles in Figs. 1(d) and 1(e)). This makes the CDW volume in 10 nm films about 1/2 the volume of 50 nm thick films[24].

It is well established that the CDW order competes with superconductivity[4]: the superconducting critical temperature $T_c$ is suppressed at the $p \approx 1/8$ doping where CDW is strongest, resulting in a deviation of the superconducting dome from the empirical parabolic shape[27,36]. As shown in various works, the CDW order also has an effect on the broadening of the resistive transition. In general, there is always a finite broadening of the resistive transition in 2-dimensional superconductors due to vortex-antivortex pair dissociation (Kosterlitz-Thouless transition)[37]. Another source of broadening is disorder, either structural (e.g., inhomogeneous doping) or electronic (e.g., CDW). In cuprates, the resistive transition is broadened around $p = 0.125$ where CDW order is strongest, and at low doping ($p < 0.08$) when approaching the AFM spin order[27,35]. Our 50 nm thin films fully exhibit

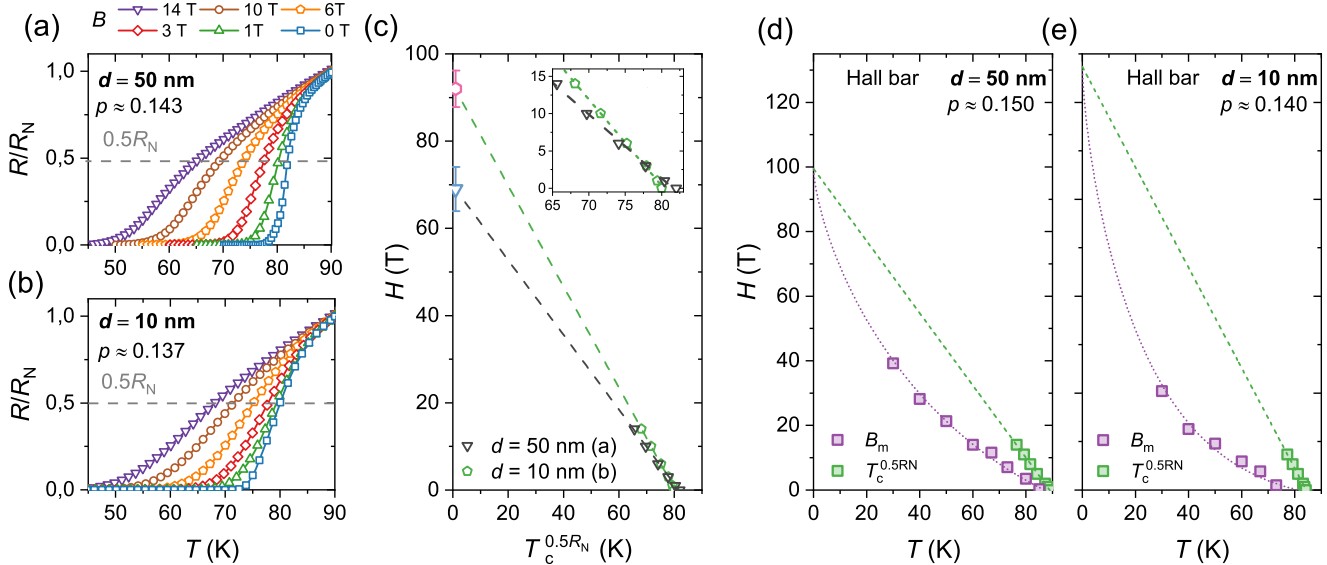

**Fig. 3 | Thickness dependence of $H_{c,2}$ in underdoped YBCO thin films. a, b** The magnetic field dependence of the resistive transition in two YBCO thin films of different thickness and similar doping level. The magnetic field is applied along the $c$-axis, perpendicular to the current which is applied along the $a$-axis. The gray dashed lines indicate where the resistance has dropped to 50% of the normal state resistance $R_N$ at $T_c$. **c,** Linear fits to $T_c^{0.5R_N}(H)$ for the two films in (**a**) and (**b**). The blue and pink symbols show the linear extrapolated values of $H_{c,2}$ at $T = 0$. The inset is a blow up of the data points to highlight the different slopes of the linear fits. The error bars are estimated from the uncertainty of the $T = 0$ value of the linear fits. **d, e** Vortex lattice melting fits for Hall bars on films with similar doping to (**a**) and (**b**). The violet squares are values of $B_M$ extracted from measurements of the resistive transition in an out-of-plane ($c$-axis) pulsed magnetic field reaching 55 T. The current is applied along the $a$-axis.

this behavior, while the 10 nm films show a transition width that is approximately twice larger than that of the 50 nm films, despite having a significantly reduced CDW volume. This suggests an intriguing outcome: removing a competing order with superconductivity results in an unexpectedly broadened resistive transition in addition to the increase in $T_c^{on}$. This effect is quite notable and does not occur in 10 nm YBCO films grown on STO (see Supplementary Fig. 2(d)), which instead show the same transition broadening as the 50 nm films at 1/8 doping. This observation points, therefore, towards a reduced superfluid stiffness in the 10 nm films. But while the increase in $T_c^{on}$ could intuitively come from the reduction of the CDW volume, a much reduced stiffness is not immediate to understand. We have resorted to our model of the $V_{eff}$ to estimate the stiffness in case of a faceted substrate (see Fig. 1(a)). The stiffness can be obtained from the imaginary part of the optical conductivity (see Supplementary section D). For the system of Fig. 1(a), corresponding to a faceted MgO substrate, we have previously demonstrated[25] that the one-dimensional faceted structure reflects in a nematic electronic structure of the coupled YBCO layers with a flattening of bands along the $a$-axis. The concomitant mass enhancement induces a significant reduction of the stiffness $D \cdot n_s/m$, with $m$ the mass and $n_s$ the superfluid density, (with respect to the homogeneous system) which explains the broader transition of the resistive transition of the 10 nm thick films. However, since only a few layers adjacent to the substrate are affected by the coupling, this only has a minor effect on the stiffness of the 50 nm films.

### Enhancement of $H_{c,2}$ in 10 nm films

In single crystals the competition between CDW and superconductivity is also seen by the rapid decrease of the zero temperature upper critical field $H_{c,2}(T = 0)$ (from now on just $H_{c,2}$) with decreasing doping.[38–42] In YBCO $H_{c,2}(p)$ has two local maxima: one in the underdoped regime ($p \approx 0.08$) and one in the overdoped regime ($p \approx 0.18$)[38], see pink dashed-dotted line Fig. 2. These doping levels do not correspond to the maximum superconducting critical temperature $T_c$, but appear to be connected to the extremes of the doping range where CDW order is present in the phase diagram[35]. Information about the

doping dependence of $H_{c,2}$ in nm thick films, with a modified CDW order, and a nematic Fermi surface could therefore shed light into the superconducting properties of the novel ground state.

In conventional low-$T_c$ superconductors $H_{c,2}(T)$ is commonly associated to $T_c^{0.5R_N}(H)$, which is defined as the temperature where the resistance has dropped to 50% of normal state value $R_N$ at the superconducting transition. This association is not straightforward in the high-$T_c$ cuprates since its temperature dependence differs from measurements of $H_{c,2}(T)$ using other methods[43]. However, it has been shown that the $T = 0$ intersect of $T_c^{0.5R_N}(H)$ can be used as an estimation of $H_{c,2}(T = 0)$[44,45]. Figure 3 shows the magnetic field dependence of the resistive transition measured in two films of similar doping ($p \approx 0.14$) but different thickness, and the linear extrapolation fits to obtain $H_{c,2}$. The magnetic field is applied along the $c$-axis (i.e. perpendicular to the $ab$ planes). We find that the field has a stronger effect on the resistive transition in the thick film, which indicates a lower value of the upper critical field compared to 10 nm thin films (see inset of Fig. 3(c)). To give further support to this result we made vortex lattice melting measurements in a 55 T pulsed field of 50 and 10 nm thick Hall bars in same range of doping as Fig. 3(a) and (b). At every temperature one sweeps the magnetic field and records the corresponding $R(H)$. The square violet points in Figs. 3(d) and (e) then correspond to the field at which the resistance is 1/100 of the value at 55 T at a defined temperature. The resulting $H(T)$ dependence can be fitted to extract the value $H_{c,2}(T = 0)$ (see Fig. 3(d) and (e)), for both 10 and 50 nm thick films, following the procedure of Ref. 40. More details about the used fitting procedure can be found in Supplementary section E. We find that the extrapolated $H_{c,2}$ values are very close to those extracted in Fig. 3(a) and (b) using the field dependent resistive transition with $H_{c,2}(T = 0)$ of 10 nm thick films larger than the one for 50 nm.

To understand the origin of this behaviour we have extracted the doping dependence of $H_{c,2}$ for the 10 nm thick films by using the same procedure as Fig. 3(a) and 3(b). Figure 4(a) shows how the slope of the linear fits decreases as the doping is reduced, indicating a decrease of $H_{c,2}$. However, the sharp decrease is offset to lower doping levels, compared to thick films and bulk crystals, which results in a strong

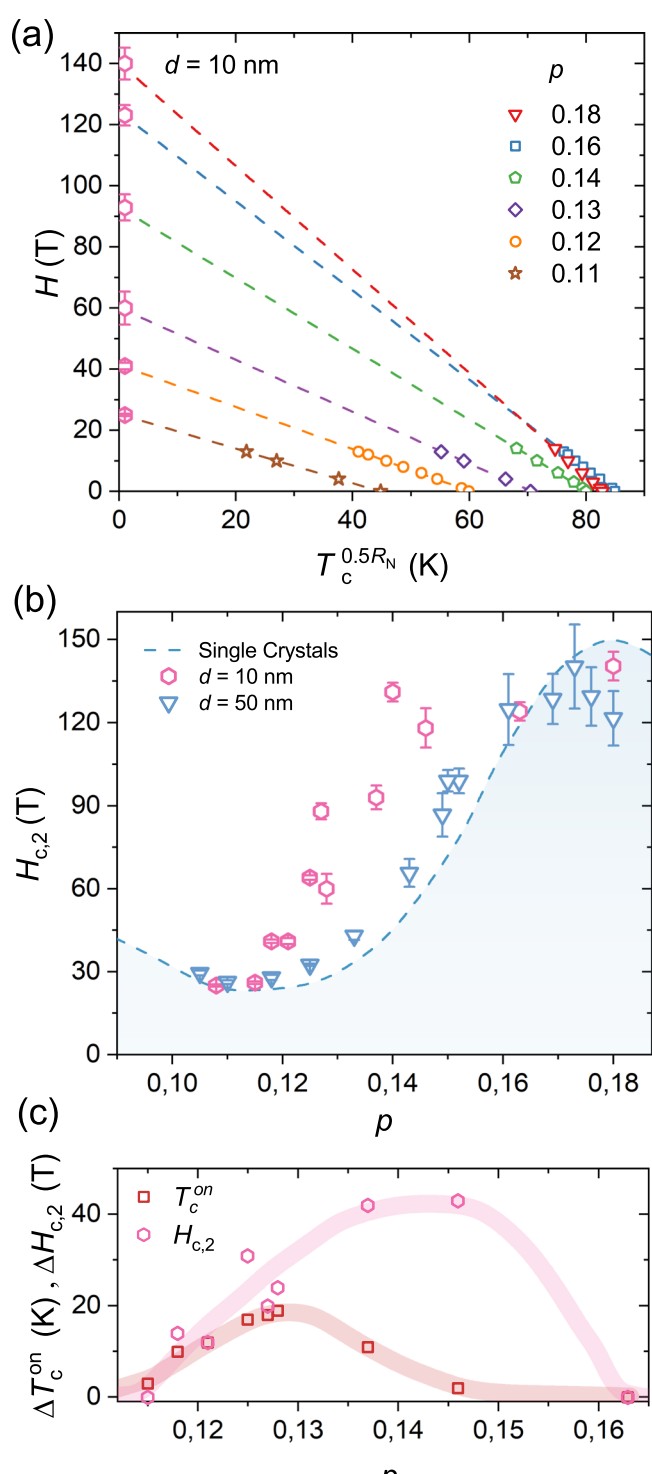

**Fig. 4 | Doping dependence of $H_{c,2}$ in strained YBCO films. a** Linear fits of $T_c^{0.5R_N}(H)$ for films with different values of $p$. The current is applied along the $a$-axis. The pink hexagons show the linear extrapolated values of $H_{c,2}$ at $T_c^{0.5R_N} = 0$. The error bars are estimated from the uncertainty of the $T=0$ value of the linear fits. **b** Doping dependence of $H_{c,2}$ in $d = 10$ nm (pink hexagons) and $d = 50$ nm (blue triangles) YBCO films. The blue dashed line shows the doping dependence of $H_{c,2}$ measured in YBCO crystals[38]. **c** Difference between $T_c^{on}$ (red squares) and $H_{c,2}$ (pink hexagons) in $d = 10$ nm and $d = 50$ nm films.

enhancement of $H_{c,2}$ in the doping range $0.12 < p < 0.16$ (see Fig. 4b). A natural question arises: is the doping range of the increase of $H_{c,2}$ in Fig. 4(b) correlated to the increase of $T_c^{on}$? Fig. 4(c) shows the doping dependence of the increase in $T_c^{on}$ and $H_{c,2}$ in the 10 nm thick films.

While the enhancement of the two quantities begins at roughly $p = 0.11$, their trend at higher doping is quite different. At $p \approx 0.145$ $\Delta T_c^{on} \approx 0$ while $\Delta H_{c,2}$ is maximized. This indicates that the enhancement for $H_{c,2}$ and $T_c^{on}$ might have different origins.

## Discussion

A nematic Fermi surface is a unique feature of our 10 nm thick films: a direct consequence of this electronic nematicity is an anisotropy of the effective masses along the $a$- and $b$-axis. As we will show below, this anisotropy is at the origin of the enhancement of $H_{c,2}$. This becomes immediately apparent from a Ginzburg-Landau (GL) analysis for a system[46,47] with the GL functional:

$$F = u(T)|\psi|^2 + \frac{1}{2}|\psi|^4 + \frac{\hbar^2}{2}\left(\frac{1}{m_a}\left|\frac{\partial\psi}{\partial x}\right|^2 + \frac{1}{m_b}\left|\frac{\partial\psi}{\partial y}\right|^2\right). \quad (1)$$

Here $\psi$ denotes the superconducting order parameter, $m_{a/b}$ are the masses along $a/b$-directions, and $u(T)$ is a function of temperature and depends on microscopic details.

The upper critical field is set by the superconducting coherence length, which is given by $\xi_{a/b}^2 = \hbar^2/(2m_{a/b}|u(T)|)$. In particular, the dependence of the upper critical field $H_{c,2}$ on the coherence length is given by $H_{c,2} \propto 1/(\xi_a\xi_b) \propto \sqrt{|m_a|}\sqrt{|m_b|}$. The masses for a nematic tight-binding model with nearest-neighbor hopping $t_{b,a} = -t(1\pm\alpha)$[24] follow the relation $m_{b,a} \sim -1/(1\pm\alpha)$. Hence, for a slight nematic distortion of an isotropic system we get $H_{c,2} \sim 1/\sqrt{1-\alpha^2}$ thus leading to a critical field $H_{c,2}$ that increases with increasing the nematicity. This simple consideration allows us to corroborate the second important finding of our paper: a Fermi surface with a mass anisotropy supports an enhancement of the upper critical field $H_{c,2}$.

To substantiate the analysis within a microscopic model, we consider the following Hamiltonian $H = H_0 + H_{CDW} + H_{SC}$ with

$$\begin{aligned}
H_0 &= \sum_{ij\sigma} t_{ij}\left(c_{i\sigma}^\dagger c_{j\sigma} + \text{H.c.}\right) - \sum_{i\sigma}\mu c_{i\sigma}^\dagger c_{i\sigma}, \\
H_{SC} &= \sum_{\langle ij\rangle}\left(\Delta_{ij}c_{i\uparrow}^\dagger c_{j\downarrow}^\dagger + \Delta_{ij}^* c_{i\downarrow}c_{j\uparrow}\right) \quad (2)\\
H_{CDW} &= \chi_{cdw}\sum_{i\sigma} e^{iQ_{cdw}\cdot r_i}c_{i\sigma}^\dagger c_{i\sigma}.
\end{aligned}$$

Here, $H_0$ describes the hopping of charge carriers on a square lattice, with hopping amplitudes $t_{ij}$ restricted to nearest- ($-t$) and next-nearest ($\sim t'$) neighbor hopping with $t'/t = -0.25$, as appropriate for bilayer cuprate SC's[48–51]. As in the above discussion of the GL argument, the nematicity of the 10 nm films is captured via a parametrization of the nearest-neighbor hopping $t_{b,a} = -t(1\pm\alpha)$[24], while the chemical potential is set to $\mu = -0.8t$ corresponding to a doping $p \approx 0.12$. Furthermore, a magnetic field $H$ is implemented in $H_0$ via the Peierls substitution.

Superconductivity is described by $H_{SC}$, where $\Delta_{ij} = -J/2\langle c_{j\downarrow}c_{i\uparrow} - c_{j\uparrow}c_{i\downarrow}\rangle$ denotes the order parameter on nearest-neighbor bonds. These are self-consistently obtained within standard BCS theory from a nearest-neighbor interaction $J$ (see Supplementary section F) and, besides the strongly prevalent $d$-wave order, we also find a small admixture of extended $s$-wave pairing for the nematic system. For the non-nematic case ($\alpha = 0$) the chosen interaction $J = 0.6t$ yields a transition temperature $T_c \approx 0.04t \approx 90$ K for $t = 200$ meV[50]. Note that the Pauli limit for cuprate superconductors is very large[52] so that the Zeeman splitting can be neglected. Moreover, the $T_c$ evaluated in our mean-field like theory neglects fluctuations and therefore should be identified with the experimental $T_c^{on}$, cf. Fig. 2.

Finally, for the CDW, we set $\chi_{cdw}$ and $Q_{cdw}$ as the CDW amplitude and wave-vectors, respectively. With numerical complexity hampering an implementation of general incommensurate CDW modulations, we approximate the CDW scattering in $H_{CDW}$ by alternative modulations that affect the Fermi surface states similarly to the experimentally

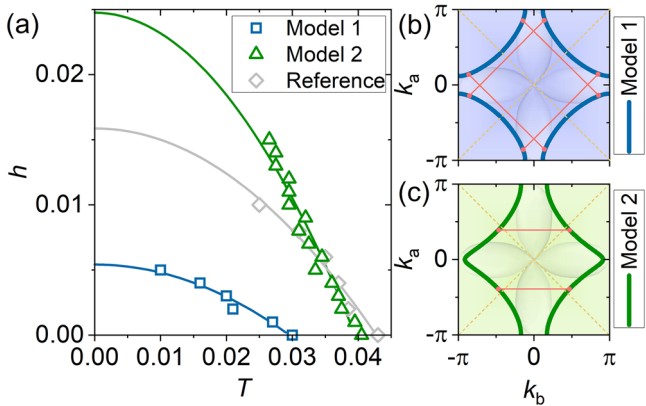

**Fig. 5 | Theoretically calculated dependence of the magnetic field on temperature. a** Magnetic field $h$ versus temperature $T$ for "Model 1" (checkerboard CDW with $Q = (\pi, \pi)$ and no nematicity $\alpha = 0$ representing 50 nm thick films, blue line and squares), "Model 2" (uniaxial CDW with $Q = (\pi, 0)$ and nematic Fermi surface $\alpha = 0.06$ representing 10 nm thick films, green line and triangles), and reference (no CDW and no nematicity $\alpha = 0$, grey line and diamonds). Solid lines are fits to a function $h = A + BT^2$, with $A$, $B$ as fit parameters. Magnetic field $h$ as reduced dimensionless field ($h = eH/(\hbar c)$) and temperature $T$ as dimensionless quantity (temperature in Kelvin given by $T \cdot t/k_B$). Fermi surfaces considered for "Model 1" (**b**) and "Model 2" (**c**). Red lines indicate CDW wave vectors, with red points marking Fermi surface points nested by these wave vectors. Yellow dashed lines represent the superconducting $d$-wave nodal lines. Since the nested points in "Model 1" are close to the antinodes, a CDW gap affects $d$-wave superconductivity more significantly than in "Model 2", where the nested points are closer to the nodes.

observed scattering. As we argue in the Supplementary section G, the two-dimensional scattering in the 50 nm films can be mimicked by a checkerboard CDW with $Q_{cdw} = (0.5, 0.5)$ [r.l.u.]. We henceforth name this model 1, see Fig. 5(b). In contrast, to model the CDW in the the 10 nm films we consider an uniaxial $Q_{cdw} = 0.5$ [r.l.u.] along the $b$-direction, as model 2, see Fig. 5(b). In both cases we set $\chi_{cdw} = 0.06t$, which for the checkerboard CDW results in $T_c \approx 0.03t \approx 70$ K, in agreement with the observed reduction in $T_c$ for the 50 nm films at $p = 0.12$ (Fig. 2). The nematic Fermi surface (model 2) yields a higher transition temperature of $T_c \approx 0.04t \approx 90$ K, also in agreement with the 10 nm films. Due to the smaller influence of CDW scattering on the nematic Fermi surface states this $T_c$ does not differ much from a reference system without CDW scattering and nematicity, see Fig. 5(a).

Upon increasing the magnetic field $H$, we identify the critical field $h_{c,2}$ (in reduced units of $h = eH/(\hbar c)$, where $H$ is the applied magnetic field) as the field where the order parameters approach $\Delta_{ij} = 0$, (see Supplementary Section F) with results plotted in Fig. 5(a). Extrapolating the extracted upper critical fields to $T = 0$ yield a significantly enhanced $h_{c,2}$ for model 2 (i.e. 10 nm films) as compared to model 1 (i.e. 50 nm films). Moreover, $h_{c,2}$ obtained for model 2 also exceeds the upper critical field of the reference model without nematicity and CDW, indicating nematicity as the primary reason of the enhancement, as also anticipated from the GL equation (1). In fact, evaluating the (hole) masses $m_{a/b} = \langle 1/(\partial^2 \xi_k/\partial k_{a/b}^2)\rangle_{FS}$, where the bracket denotes the FS average and $\xi_k$ is the band dispersion, we find (in units of $t^{-1}$): $m_a = m_b = -0.62$ for the non-nematic reference state and $m_a = -0.31$, $m_b = -2.16$ in case of model 2. GL theory Eq. (1) then results in $h_{c,2}^{m2}/h_{c,2}^{ref} = 1.32$ whereas the microscopic results in Fig. 5(a) yields $h_{c,2}^{m2}/h_{c,2}^{ref} = 1.56$. This allows us to conclude that the nematic ground state is indeed the dominant source for the enhancement of $h_{c,2}$ in the 10 nm films. It is also important to note that the opening of a pseudogap does not influence this conclusion. In general, the mass anisotropy is only weakly dependent on the Fermi surface momenta so that also the nodal parts of the (nematic) Fermi surface contribute significantly to the $h_{c,2}$ enhancement.

This allows us to summarize our findings as follows: the thin YBCO films grown on MgO are subject to a one-dimensional faceted substrate structure, which induces a nematic electronic state in the superconducting film. The concomitant modification of the two-dimensional CDW into a one-dimensional CDW in the doping range $0.12 < p < 0.15$ leads to an enhancement of $T_c$, likely due to the reduced Fermi surface area affected by CDW scattering (see Fig. 5(b)–(c)). However, the CDW component along the $b$-axis-which competes with superconductivity-persists, so we cannot exclude a possible additional contribution from band flattening to the observed $T_c$ enhancement[53]. On the other hand, the increase in $H_{c,2}$ beyond the bulk value is driven by the nematic character of the electronic structure and cannot be solely attributed to the formation of a one-dimensional CDW. In this regard, we have also verified that the inclusion of additional anisotropy in the pairing interaction yields only a negligible contribution to the observed $H_{c,2}$ enhancement. This reinforces the generality of our finding: the enhancement of the $H_{c,2}$ is a mechanism fully driven by the Fermi surface distortion and can therefore be tuned by properly designing the band structure of the cuprates through substrate nanoengineering. Ultimately, our results demonstrate that the concept of band engineering by using a peculiar substrate morphology can strongly modify the ground state of the material and could be extended to other strongly correlated systems, opening new avenues for exploring exotic quantum phases and phenomena.

## Data availability
All data shown in the main text and in the supplementary information are available at the Zenodo repository[54], under accession code https://doi.org/10.5281/zenodo.17578632.

## Code availability
The theoretical analysis was carried out with FORTRAN codes to implement numerical integrations for solving the gap equations, calculating the stiffness and obtaining the critical fields reported in Fig. 5. The FORTRAN codes are available from two of the authors [D.C., G.S.] upon request.

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

## Acknowledgements

We acknowledge founding support by the Swedish Research Council (VR) under the projects 2020-04945 (R.A.), 2020-05184 (T.B.), 2022-04334 (F.L.) and 2022-03963 (A.B.S.), by the Knut and Alice Wallenberg foundation under the projects KAW 2023-0341 (F.L.) and KAW 2024-0129 (F.L. and A.B.S.), by the EIC pathfinder grant

from the European Union 101130224 'JOSEPHINE' (F.L., T.B.), the COST action 'SUPERQUMAP' (F.L.) and by the Deutsche Forschungsgemeinschaft, under SE 806/20-1 (G.S.). This work was performed in part at Myfab Chalmers.

## Author contributions

F.L. and E.W. conceived the experiments with suggestions from R.A., T.B. and G.S. E.W. grew the films and characterized the transport properties at low magnetic fields with support from R.A. and A.K. R.A., E.W. T.B. and F.L. performed the RIXS measurements. R.A. analysed the RIXS experimental data. D.C., A.B.S. and G.S. developed the modelling and the theory analysis of the experiment. The high magnetic field measurements were performed by D.W., C.P., E.W., R.A., T.B. and F.L. All the authors discussed and interpreted the data. F.L, E.W., D.C., R.A. A.B.S. and G.S. wrote the manuscript with contributions from all authors.

## Funding

## Competing interests

The authors declare no competing financial interests.
