## [Transparent Peer Review file · Nature Communications]

Boosting Superconductivity in ultrathin YBa₂Cu₃O_{7- δ} films via nanofaceted substrates

Corresponding Author: Professor Floriana Lombardi

Version 0:

Reviewer comments:

Reviewer #1

(Remarks to the Author)

The authors in the manuscript “Boosting superconductivity: how nanofaceted surfaces transform the ground state of ultrathin YBa₂Cu₃O_{7- δ} thin films” performed the study of the superconducting properties of the YBCO thin films with different thicknesses and doping level. They observed a significant increase in the critical temperature and upper magnetic field for underdoped 10 nm thick films compared to 50 nm thick films with the same doping level. The authors concluded that the difference in the superconducting properties between the 10-nm- and 50-nm-thick films is driven by electronic nematicity and unidirectional charge density waves, and proposed a theoretical model explaining the difference in T_c and H_{c2} for the 10-nm- and 50-nm-thick films.

I have no questions about the quality and completeness of the experimental data presented in the manuscript. However, I have a few comments regarding the interpretation of the experimental data and some experimental details.

1. The authors wrote that “We observe that the slope of the $\rho(T)$ is very different between the a and the b directions, and that the resistivity anisotropy ρ_a/ρ_b is strongly enhanced even at room temperature. These properties can be accounted for by considering that the Fermi surface becomes nematic in very thin films [22].” According to the TEM and SEM images presented in previous publication of the authors cited in Ref. 26 and 27, the YBCO film grown on the (110) MgO substrate has elongated grains, structural mosaicity (CuO₂ planes are tilted with respect to the substrate surface), and crystallographic defects. The film structure rather than the nematic Fermi surface may be responsible for the large in-plane anisotropy in such films.

2. Do the authors have AFM images of YBCO films on (110) MgO substrate? They might shed some light on the origin of the in-plane anisotropy.

3. The authors used the two different approaches to estimate the doping level of the YBCO films. One approach is based on the measurement of the c-axis length and T_c. The other approach uses the pseudogap temperature obtained from the R(T) curve nonlinearity to determine the doping level.

According to the previous publication of the authors cited in Ref. 26 (Fig. 6), the dependence of T_c on the c-axis length of YBCO film is not single-valued. YBCO films deposited on (100) SrTiO₃ and (110) MgO substrates have different c-axis values despite the same T_c value which rises the question how to compare the doping level of the films deposited on different substrates.

The use of pseudogap temperature dependence to determine the doping level of thick and nm-thick films on SrTiO₃ substrate is reasonable. In this case, the authors refer on their previous study with 30-nm-thick YBCO films cited as Ref. 26. However, the determination of the doping level from the pseudogap temperature for a 10-nm-thick film on a MgO substrate is, in my opinion, questionable.

There is a large lattice constant mismatch between YBCO and MgO. The strain in heterostructures with the large lattice constant mismatch can be relaxed in a thin layer (around 10 nm) due to crystallographic defects. This is confirmed by Fig. 7d in previous publication of the authors cited as Ref. 27. Therefore, the resistivity of the nm-thick film should consist of two components: intra-grain resistivity and resistivity component due to different crystallographic defects. This assumption agrees well with the R(T) curves presented by the authors. The temperature of the R(T) curve slope change are the same for the 10-nm and 50-nm-thick films on (100) SrTiO₃ substrates due to the small lattice constant mismatch and consequently small number of crystallographic defects in the 10-nm-thick films. But the temperature of the R(T) curve slope change can be different for the 10-nm and 50-nm-thick films on (110) MgO substrates because of the large number of crystallographic defects in the 10-nm-thick film. The procedure for determining the doping level from the change in the slope of the R(T) curve is valid only under the assumption that the resistivity contribution due to various crystallographic defects is small or

independent of temperature, which was not justified in the manuscript.

As the evaluation of the doping level of YBCO thin films is the cornerstone of the manuscript, it should be revised or better explained to take into account the different structure and strain of films deposited on SrTiO₃ and MgO substrates.

4. It is known that pressure and strain affect the superconducting properties of cuprate superconductors. For example, the critical temperature of LSCO can be increased by 50% due to strain [1]. The critical temperature of YBCO also depends on the pressure or strain induced by the substrate [2]. The effects of pressure and strain can significantly increase the T_c of underdoped YBCO films, while the T_c of optimally and overdoped YBCO films are little or negatively affected [3], which is in agreement with the authors' findings for the 10-nm-thick films that are strained. The change in T_c of YBCO was explained by strain and pressure induced doping [3]. The strain-induced doping may be at least partly responsible for the difference in T_c between 10 and 50 nm thick YBCO films on a (110) MgO substrate, and the authors should discuss these effects in the manuscript to get a complete picture.

5. Did the surface of the SrTiO₃ substrate used in this work have SrO or TiO₂ termination?

6. Were the oxygen annealing conditions the same for 10 nm and 50 nm thick films with the same doping level as it was determined in the manuscript?

1. Zhang et al. A Review on Strain Study of Cuprate Superconductors. *Nanomaterials* 2022, 12, 3340.

2. Zhai et al., Effect of interfacial strain on critical temperature of YBa₂Cu₃O_{7-δ} thin films, *APPL. PHYS. LETT.* 76(23), 3469 (2000)

3. Jurkutat et al., How pressure enhances the critical temperature of superconductivity in YBa₂Cu₃O_{6+y}, *PNAS* 2023 Vol. 120 No. 2 e2215458120

Reviewer #2

(Remarks to the Author)

The authors present experimental results on strain engineering for the cuprate high-T_c superconductor YBCO. In particular, they address how substrate-induced strain alters the electronic properties. Here, strain is applied via a MgO substrate on which thin films of YBCO are grown. The use of MgO leads to uniaxial strain, which induces in turn a nematic deformation of the Fermi surface. Moreover, the nano-faceted texture of the substrate yields through random coupling to the thin film, an effective potential on electrons in the YBCO film. The uniaxial strain also induces a unidirectional charge density wave (CDW) order, in contrast to the bi-directional CDW usually observed in unstrained samples.

The key finding is that this uniaxial strain leads to an enhancement of both the superconducting onset temperature and the upper critical field within a specific doping range. To systematically investigate this effect, the authors compare YBCO films of different thicknesses. The 10 nm film exhibits the strain-induced features prominently, while the 50 nm film serves as a reference system, close to bulk-like behavior.

Additional theoretical analysis is provided to support the interpretation of the experimental observations, offering a plausible explanation for the observed enhancement of superconductivity under strain. Overall, the results are both exciting and, to my best knowledge, novel. The manuscript presents a substantial body of data and a discussion covering a wide range of arguments to support its conclusions. However, the complexity of the discussion may make it challenging for readers unfamiliar with the field to grasp the central message. To improve the clarity and accessibility of the manuscript, I would like to make the following suggestions and add also a few comments:

1) The manuscript would benefit from a clear and concise conclusion explaining the primary mechanism responsible for the observed enhancement of superconductivity, in the view of the authors. In particular, the role of unidirectional CDW order and disorder—both of which are discussed—remains ambiguous. A focused summary explicitly stating whether the enhancement is driven by Fermi surface deformation, modified CDW interactions, or disorder-induced effects would strengthen the overall argument.

2) The theoretical discussion centers primarily on band structure modifications under strain. However, since superconductivity in cuprates arises from electron-electron interactions leading to Cooper pairing, it is natural to ask whether strain also affects the pairing interaction itself. In my view it would be in order to at least comment on why such effects might be negligible or how they could reinforce the observed phenomena.

3) YBCO is already intrinsically not tetragonal, but orthorhombic (CuO chains). Does this play a role for the strain engineering?

4) The enhancement of the upper critical field is attributed to anisotropic effective masses in the Ginzburg-Landau framework, specifically through the relationship $m_a/b = m \pm \delta m$. While this leads to reduced in-plane coherence lengths, the assumption that the average mass remains constant seems critical—and potentially unrealistic. Later in the manuscript (in Model 2 on p. 7), this condition is not fulfilled, which undermines the robustness of the GL-based argument. This discrepancy should be addressed explicitly, or the argument should be reformulated to reflect the more general case.

5) On page 2, the authors mention that substrate-induced randomness can be treated using the coherent potential approximation (CPA). However, it is unclear whether or where this approach is applied later in the manuscript. It may be embedded in the model presented in the Supplementary Material (Eq. S1), but this is not made explicit. The interpretation of the effective potential term V_{eff} , likely corresponding to the second term in Eq. (S1), is also unclear. A brief explanation of the origin and physical meaning of this term—particularly its functional form—would help the reader follow the theoretical modelling more easily.

6) One of the strengths of the study is the controlled comparison between strained and unstrained systems across different doping levels. The manuscript references the “empirical parabolic T_c(p)” relation, which appears to play an important role in the analysis, especially in connection to Figure 2 and the red-shaded region. However, this concept is not sufficiently explained. A brief discussion—perhaps in the Supplementary Material—detailing the origin and functional form of the parabolic relation, and how it is used to extract doping levels, would greatly enhance the clarity of this comparison.

In summary, I believe that this manuscript covers material which deserves publication in *Nature Communications*. However,

the manuscript should be made more easily accessible to the non-expert reader.

Version 1:

Reviewer comments:

Reviewer #1

(Remarks to the Author)

I appreciate the authors' efforts to address all of my comments. Having read their responses and the revised manuscript, I believe it deserves publication in Nature Communications.

Reviewer #2

(Remarks to the Author)

The authors have addressed the comments from both reviewers and have refined their manuscript accordingly. In particular, their responses effectively clarify the issues raised in my previous comments. I also followed the insightful comments provided by Reviewer #1, and I find the authors' replies to those points satisfactory as well. Although some experimental aspects are rather complex, the author's clarifications provide sufficient justification to present these results and ideas to the research community. Therefore, I have no reservations to recommend the publication of the manuscript in its present form.

In the following, we address each of the referees' questions and comments point by point.

Reviewer #1 (Remarks to the Author):

The authors in the manuscript "Boosting superconductivity: how nanofaceted surfaces transform the ground state of ultrathin $\text{YBa}_2\text{Cu}_3\text{O}_{7-\delta}$ thin films" performed the study of the superconducting properties of the YBCO thin films with different thicknesses and doping level. They observed a significant increase in the critical temperature and upper magnetic field for underdoped 10 nm thick films compared to 50 nm thick films with the same doping level. The authors concluded that the difference in the superconducting properties between the 10-nm- and 50-nm-thick films is driven by electronic nematicity and unidirectional charge density waves, and proposed a theoretical model explaining the difference in T_c and H_{c2} for the 10-nm- and 50-nm-thick films.

I have no questions about the quality and completeness of the experimental data presented in the manuscript. However, I have a few comments regarding the interpretation of the experimental data and some experimental details.

We sincerely appreciate the referee for the positive assessment of our manuscript and we hope our responses will meet the referee's expectations.

1. The authors wrote that "We observe that the slope of the $\rho(T)$ is very different between the a and the b directions, and that the resistivity anisotropy ρ_a/ρ_b is strongly enhanced even at room temperature. These properties can be accounted for by considering that the Fermi surface becomes nematic in very thin films [22]."

According to the TEM and SEM images presented in previous publication of the authors cited in Ref. 26 and 27, the YBCO film grown on the (110) MgO substrate has elongated grains, structural mosaicity (CuO₂ planes are tilted with respect to the substrate surface), and crystallographic defects. The film structure rather than the nematic Fermi surface may be responsible for the large in-plane anisotropy in such films.

The referee is correct that the substrate (not the film) has elongated nanofacets and the films present buckling (see Fig. 7 of our Phys. Rev. Materials 3, 114804 (2019)). However we exclude that the phenomenology we observe, and in particular the anisotropy of the resistivity along the a-axis and b-axis, can be attributed to defects. The 10-nm films grown on both MgO and STO show essentially the same average resistivity ρ_{aver} at room temperature, as shown in Figure R1 below, indicating comparable defect scattering across various substrates. In addition, when the linear high-temperature part of $\rho(T)$ is extrapolated to $T \rightarrow 0$, the intercepts ρ_{0a} and ρ_{0b} are also quite comparable, showing that the residual resistivity and thus defect contribution is rather isotropic. One needs also to consider that our 10 nm films show a strong

Figure R1: Restivity as a function of the temperature for our 10 nm thick YBCO films grown on (a) MgO (110) and (b) STO (001) substrates. Blue and pink curves refer to a- and b- axis, respectively.

elongation of the b-axis (about 2%) compared to the 50 nm thick YBCO [Science 373, 1506 (2021), i.e., Ref. 24 of the main text], while the a-axis is almost unchanged. The increase in b-axis length effectively reduces the hopping parameter t in the CuO_2 planes along the b-direction which in turns should increase ρ_b

making it closer to ρ_a . Instead we observe exactly the opposite effect. These considerations demonstrate that the anisotropy is intrinsic in origin rather than defect-related: even considering some sample-to-sample variations of $\rho_{0a,b}$ and ρ_{aver} (on the order of 10–20%) we cannot explain the large effect in transport we observe as due to defects. Finally, it is known that defects can strongly affect the d-wave order parameter in YBCO acting as depairing mechanism. Instead we observe that the T_c onset of 10 nm films on MgO is largely enhanced compared to 50 nm thick films and to 10 nm films grown on STO. Overall these facts strongly support the claims of our paper.

2. Do the authors have AFM images of YBCO films on (110) MgO substrate? They might shed some light on the origin of the in-plane anisotropy.

Yes, we do have AFM (and SEM) images of 10 nm and 50 nm thick (untwinned) YBCO films on (110) MgO substrates, see Figs. R2 and R3 below.

Figure R2: (a) SEM and (b) AFM pictures of our 10 nm thick YBCO films on MgO (110). In panel (b), R_q is the average surface roughness evaluated on the whole $5 \times 5 \mu\text{m}^2$ area.

Figure R3: (a) SEM and (b) AFM pictures of our 50 nm thick YBCO films on MgO (110). In panel (b), R_q is the average surface roughness evaluated on the whole $5 \times 5 \mu\text{m}^2$ area, which is comparable to that of the thinnest films.

There are shallow channels running along $[1-10]$ MgO (YBCO a-axis) in both 10 nm and 50 nm thick films and the average surface roughness (R_q) is nearly the same. The white spots in the SEM of the 10 nm thick film come from charging of the insulating substrate. We do not believe that the film structure can be responsible for the anisotropic transport: 1) the channels are shallow around 1-2 nm deep and 2) they run along the a-axis so they could eventually act as scattering centers in the perpendicular direction, i.e., the b-axis increasing its resistivity. Instead we observe the opposite effect; 3) the morphology of the 10 nm and 50 nm are very comparable.

3. The authors used the two different approaches to estimate the doping level of the YBCO films. One approach is based on the measurement of the c-axis length and T_c . The other approach uses the pseudogap temperature obtained from the $R(T)$ curve nonlinearity to determine the doping level. According to the previous publication of the authors cited in Ref. 26 (Fig. 6), the dependence of T_c on the c-axis length of YBCO film is not single-valued. YBCO films deposited on (100) SrTiO₃ and (110) MgO substrates have different c-axis values despite the same T_c value which is rise the question how to compare the doping level of the films deposited on different substrates.

The use of pseudogap temperature dependence to determine the doping level of thick and nm-thick films on SrTiO₃ substrate is reasonable. In this case, the authors refer on their previous study with 30-nm-thick YBCO films cited as Ref. 26. However, the determination of the doping level from the pseudogap temperature for a 10-nm-thick film on a MgO substrate is, in my opinion, questionable. There is a large lattice constant mismatch between YBCO and MgO. The strain in heterostructures with the large lattice constant mismatch can be relaxed in a thin layer (around 10 nm) due to crystallographic defects. This is confirmed by Fig. 7d in previous publication of the authors cited as Ref. 27. Therefore, the resistivity of the nm-thick film should consist of two components: intra-grain resistivity and resistivity component due to different crystallographic defects. This assumption agrees well with the $R(T)$ curves presented by the authors. The temperature of the $R(T)$ curve slope change are the same for the 10-nm and 50-nm-thick films on (100) SrTiO₃ substrates due to the small lattice constant mismatch and consequently small number of crystallographic defects in the 10-nm-thick films. But the temperature of the $R(T)$ curve slope change can be different for the 10-nm and 50-nm-thick

films on (110) MgO substrates because of the large number of crystallographic defects in the 10-nm-thick film.

We thank the referee for their comment. In the previous answers we believe we have convincingly excluded that the transport phenomenology of 10 nm films on MgO is due to defects.

The procedure for determining the doping level from the change in the slope of the $R(T)$ curve is valid only under the assumption that the resistivity contribution due to various crystallographic defects is small or independent of temperature, which was not justified in the manuscript.

As the evaluation of the doping level of YBCO thin films is the cornerstone of the manuscript, it should be revised or better explained to take into account the different structure and strain of films deposited on SrTiO₃ and MgO substrates.

We agree that a stronger justification for why the same procedure for determining the doping can be used on films of different thickness is important for the results of the paper and we thank the referee for the comment.

The reason that we think that we can use the pseudogap temperature as an estimate of the doping level on the 10 nm thick films on MgO, despite the large lattice mismatch, builds on the following:

i) If we compare the $R(T)$ curves of 10 nm and 50 nm thick films on MgO, we find that along the b -axis the $R(T)$ curve slope change depends on the thickness, as the referee correctly states. However, if we consider the transport along the a -axis, the $R(T)$ curve slope change is the same (within a few K), independent of the film thickness. If the curve slope change would be a result of crystallographic defects, they would have to act in a one-dimensional way (only along the b -axis), which we do not think is plausible. There is no evidence in our structural measurements (i.e. AFM, SEM, XRD) that supports this scenario. In addition, the residual resistivity ρ_0 is comparable along the a -axis and b -axis. Therefore we are convinced that the a -axis $R(T)$ slope change can be used as an estimate of the doping level in the 10 and 50 nm thick films.

ii) However, it is worth to point out that, despite the reasoning outlined above, the use of the pseudogap temperature to estimate the doping level should be regarded as a cross-check—rather than the primary method—within our analysis. Our standard procedure relies on the determination of the c -axis lattice parameter and the agreement of the measured T_c with the empirical parabolic $T_c(p)$ relation at both low and high doping levels. The pseudogap-based estimation provides additional validation for this approach. Prompted also by the valuable comments from the second referee, we have added a new section - now Section 1 - at the beginning of the Supplementary Material, where we explicitly discuss the doping determination procedures we have followed for our films.

4. It is known that pressure and strain affect the superconducting properties of cuprate superconductors. For example, the critical temperature of LSCO can be increased by 50% due to strain [1]. The critical temperature of YBCO also depends on the pressure or strain induced by the substrate [2]. The effects of pressure and strain can significantly increase the T_c of underdoped YBCO films, while the T_c of optimally and overdoped YBCO films are little or negatively affected [3], which is in agreement with the authors' findings for the 10-nm-thick films that are strained. The change in T_c of YBCO was explained by strain and pressure induced doping [3]. The strain-induced doping may be at least partly responsible for the difference in T_c between 10 and 50 nm thick YBCO films on a (110) MgO substrate, and the authors should discuss these effects in the manuscript to get a complete picture.

We thank the referee for raising this important point and for prompting us to clarify the distinction between hydrostatic pressure, mentioned by the referee and causing an increased

critical temperature in the underdoped region, and the specific type of strain applied in our experiment. This clarification is indeed essential to avoid any misunderstanding in the interpretation of our results.

We agree that hydrostatic pressure and strain can significantly influence the superconducting properties of cuprates. However, as we now emphasize in the revised manuscript, the enhancement of T_c observed in our 10 nm-thick films cannot be explained by a direct analogy with hydrostatic pressure experiments, where the unit cell is uniformly compressed in all directions. In contrast, our films are subjected to a uniaxial tensile strain, imposed by the anisotropic epitaxial interface with the nanofaceted substrate. This condition is fundamentally different, both in symmetry and in physical effect, from hydrostatic compression. In the framework of previous hydrostatic pressure experiments, we should not be able to see sizable enhancement of T_c .

In addition, the change in lattice constants when reducing the film thickness from 50 nm to 10 nm is -0,4% in c and +0,4% in b (while a is unchanged, being the strain in our films unidirectional). These lattice modifications correspond to pressures of about 0.5 GPa on the c -axis and 1 GPa on the b -axis [Phys. Rev. Lett. 78, 1960 (1997)]. In previous studies on YBCO [Appl. Phys. Lett. 76, 3469 (2000); Proc. Natl. Acad. Sci. U.S.A. 120, e2215458120 (2023)], including the references of the referee [Nanomaterials 12, 3340 (2022)], it has been shown that these levels of strain results in changes of T_c of maximum a few K, which is much less than the effects we observe in our study. Therefore, any effect of doping changes (if present) due to strain would be negligible.

We do however agree that a discussion about the effects of strain on doping would improve the manuscript. Therefore, to make this point clearer to the reader, we have added a dedicated paragraph in the Results section of the revised manuscript (see page 3, “Here, it is worth emphasizing...”).

Finally, we have added the relevant review by Zhang et al., [Nanomaterials 12, 3340 (2022)]—mentioned by the referee—as a new reference in the Introduction (now Ref. [15]), to better frame the broader context of strain effects in cuprate superconductors.

5. Did the surface of the SrTiO₃ substrate used in this work have SrO or TiO₂ termination?

We did not apply any specific chemical treatment (e.g., by HF etching) to achieve a single termination on our SrTiO₃ substrates. As a result, the substrates used in this work exhibit a mixed termination, with both SrO and TiO₂ terminations present. Importantly, we have previously optimized the growth of YBCO films on such mixed-terminated SrTiO₃ substrates, as reported in Phys. Rev. Materials 2, 024804 (2018). Films grown under these conditions consistently showed optimal structural and transport properties. We have added this information to the main text, which now reads: “For comparison we have also deposited 10 nm and 50 nm thick YBCO (close to the 1/8 doping) on double-terminated (SrO and TiO₂) SrTiO₃ (STO) substrates”.

6. Were the oxygen annealing conditions the same for 10 nm and 50 nm thick films with the same doping level as it was determined in the manuscript?

Yes, the annealing conditions were nominally the same for both 10 nm and 50 nm thick films. However, we have observed that, over time, the same annealing pressure may result in different doping levels due to gradual changes over time in the PLD deposition parameters. For this reason, we regularly recalibrate the oxygen annealing parameters to ensure consistent doping levels. This ongoing calibration process was applied uniformly to both thicknesses.

Reviewer #2 (Remarks to the Author):

Overall, the results are both exciting and, to my best knowledge, novel. The manuscript presents a substantial body of data and a discussion covering a wide range of arguments to support its conclusions. However, the complexity of the discussion may make it challenging for readers unfamiliar with the field to grasp the central message. To improve the clarity and accessibility of the manuscript, I would like to make the following suggestions and add also a few comments:

We thank the referee for the positive and encouraging assessment of our work. We appreciate the suggestion to improve the clarity and accessibility of the manuscript. In the revised version, we have carefully revised the text to streamline the discussion and better highlight the central message of the paper. We hope these changes address the referee's concerns and improve the overall readability of the manuscript.

1) The manuscript would benefit from a clear and concise conclusion explaining the primary mechanism responsible for the observed enhancement of superconductivity, in the view of the authors. In particular, the role of unidirectional CDW order and disorder—both of which are discussed—remains ambiguous. A focused summary explicitly stating whether the enhancement is driven by Fermi surface deformation, modified CDW interactions, or disorder-induced effects would strengthen the overall argument.

We have expanded the last paragraph in the revised version, page 7, where the points mentioned by the referee are explicitly summarized. In particular, we now clearly state that the T_c enhancement is induced by the modification of the CDW from 2D to 1D which reduces the concomitant suppression of the density of states at the Fermi energy. At the same time the fact that the T_c enhancement goes beyond the theoretical parabolic dome close to 1/8 doping tells that effect of band flattening and enhanced superconductivity might play a role. On the other hand, we want to emphasize the enhancement of $H_{c,2}$ is due to the nematic electronic structure and not primarily induced by the 1D CDW.

2) The theoretical discussion centers primarily on band structure modifications under strain. However, since superconductivity in cuprates arises from electron-electron interactions leading to Cooper pairing, it is natural to ask whether strain also affects the pairing interaction itself. In my view it would be in order to at least comment on why such effects might be negligible or how they could reinforce the observed phenomena.

We appreciate this comment of the referee. Of course, there is no consensus on the microscopic origin of the pairing mechanism in cuprates yet. However, let's assume for example that it is due to the exchange of spin-fluctuations. The spin-spin coupling constant $J = \frac{4t^2}{U}$ can be related to an exchange process which involves a doubly occupied site with energy U . In a nematic system it becomes anisotropic with $J_a = \frac{4t_a^2}{U} = \frac{4t^2}{U} (1 - \alpha)^2$ and $J_b = \frac{4t_b^2}{U} = \frac{4t^2}{U} (1 + \alpha)^2$, where we have used the same parametrization of the hopping integrals as in our manuscript. For the nematic system we then find that inclusion of the anisotropy in the coupling changes the dominant d-wave order parameter by less than 1%. From a BCS perspective, where the coherence length is given by $\xi = \hbar v_F / \Delta$ this means that the anisotropy in the pairing interaction only has a negligible influence on ξ and therefore on the critical field $H_{c,2}$. Thus, the anisotropy in the band structure under strain, via the mass (or v_F) anisotropy, is the dominating ingredient to have an increase of $H_{c,2}$. We comment on this in the last paragraph of our revised manuscript in the Discussion section.

3) YBCO is already intrinsically not tetragonal, but orthorhombic (CuO chains). Does this play a role for the strain engineering?

We thank the referee for raising this important point. The presence of CuO chains along the b-axis give rise to an in-plane anisotropy in the absolute resistivity values and in the slopes of $\rho(T)$. These differences depends on doping and are maximum in fully oxygenated films. However, in thin films, the long-range ordering of the chains is lower than single crystals—even in detwinned samples—and the resulting anisotropy becomes negligible in particular at low doping. Indeed, for 50 nm-thick YBCO films grown on (110) MgO, we find that the in-plane resistivity anisotropy at room temperature between the a- and b-axes is relatively small - on the order of 20% (see Fig. 1b in the main text) and the difference in slopes almost negligible.. In addition the temperature dependence of the resistivity is very similar along the two directions, suggesting that the presence of CuO chains primarily enhances conductivity along the b-axis, without significantly affecting its temperature dependence (at least above the superconducting transition).

Consequently, we believe that the strain engineering approach we propose could be effective even in cuprate compounds without intrinsic orthorhombicity, such as LSCO.

4) The enhancement of the upper critical field is attributed to anisotropic effective masses in the Ginzburg-Landau framework, specifically through the relationship $m_a/b = m \pm \delta m$. While this leads to reduced in-plane coherence lengths, the assumption that the average mass remains constant seems critical—and potentially unrealistic. Later in the manuscript (in Model 2 on p. 7), this condition is not fulfilled, which undermines the robustness of the GL-based argument. This discrepancy should be addressed explicitly, or the argument should be reformulated to reflect the more general case.

We thank the referee for pointing out this issue. In fact, the presentation of this point was not correct in our previous version. In the revised version we relate the anisotropy of the masses to the anisotropy of the hopping parameters in the microscopic model. This obviously leads to a geometric mean of the anisotropic masses which is larger than the mass of the isotropic system, consistent with what has also been realized by the referee.

5) On page 2, the authors mention that substrate-induced randomness can be treated using the coherent potential approximation (CPA). However, it is unclear whether or where this approach is applied later in the manuscript. It may be embedded in the model presented in the Supplementary Material (Eq. S1), but this is not made explicit. The interpretation of the effective potential term V_{eff} , likely corresponding to the second term in Eq. (S1), is also unclear. A brief explanation of the origin and physical meaning of this term—particularly its functional form—would help the reader follow the theoretical modelling more easily.

We thank the referee for pointing out the ambiguities related to the presentation of our model and the role of the coherent potential approximation (CPA). In our previous paper (Ref. 25), we proposed a microscopic model to account for the different behaviors observed in transport experiments on thin cuprate films grown on MgO (110) and STO (001) substrates. The essential difference arises from the distinct surface morphologies of the two substrates: while the 1° vicinity of the STO surface leads to step edges along the b-axis with atomic-scale height, the MgO surface is characterized by nanofacets, also oriented along the b-axis, but with significantly larger height and extension. Both types of surface features interact with the cuprate film via unsaturated bonds, but the effective potential generated by virtual hopping processes of charge carriers between film and substrate is weaker in the case of STO. In Ref. 25, the randomness associated with both the step edges and facets was incorporated via the CPA, which successfully reproduced the emergence of a nematic electronic structure. In that study (as well as in Ref. 24), the nematicity was introduced phenomenologically as an

anisotropy in the hopping amplitudes along the a- and b-directions—a scheme we also adopt in the present work [cf. Eq. (2) and the discussion that follows].

For the calculation of the stiffness, presented in Supplementary Section D, we employ a microscopic model similar to that used in Ref. 25 and now include a brief summary of the model, along with the explicit form of the effective potential. However, in this case we do not use the CPA. Instead, we evaluate the stiffness over several strip configurations and perform a direct averaging. Although this approach is limited to finite lattice sizes (up to 40×40), it has the advantage of treating disorder exactly.

Since the CPA is not employed in the present manuscript, we have removed the corresponding mention from the Introduction, which we believe was the source of the confusion.

6) One of the strengths of the study is the controlled comparison between strained and unstrained systems across different doping levels. The manuscript references the “empirical parabolic $T_c(p)$ ” relation, which appears to play an important role in the analysis, especially in connection to Figure 2 and the red-shaded region. However, this concept is not sufficiently explained. A brief discussion—perhaps in the Supplementary Material—detailing the origin and functional form of the parabolic relation, and how it is used to extract doping levels, would greatly enhance the clarity of this comparison.

We thank the referee for highlighting the importance of clarifying the role of the empirical parabolic $T_c(p)$ relationship in our analysis. Although this analytical expression was originally established for LSCO in the early days of high- T_c superconductivity, and we have discussed its relevance to estimate the doping of our thin films in a previous work [Phys. Rev. Materials 2, 024804 (2018)], we agree that in the present study the determination of doping plays a sufficiently central role to warrant including at least the basic background in the manuscript. Following the referee’s suggestion, we have therefore added a dedicated paragraph in the Supplementary Material explaining the origin, functional form, and limitations of the parabolic $T_c(p)$ relation. We also explain how this relation, combined with structural parameters such as the c-axis lattice constant, is used to estimate doping levels across our series of strained and unstrained films. We hope this addition enhances the clarity and transparency of the comparison presented in Figure 2 of the main text.

In summary, I believe that this manuscript covers material which deserves publication in Nature Communications. However, the manuscript should be made more easily accessible to the non-expert reader.

Again, we are grateful for the referee’s positive overall evaluation and for recognizing the significance of our work. We believe that the manuscript has improved thanks to their suggestions, and we sincerely hope the referee finds the modifications satisfactory.

Yours sincerely

Götz Seibold and Floriana Lombardi